# Perfusion Status in Lacunar Stroke: A Pathophysiological Issue

**DOI:** 10.3390/diagnostics13122003

**Published:** 2023-06-08

**Authors:** Marialuisa Zedde, Manuela Napoli, Ilaria Grisendi, Federica Assenza, Claudio Moratti, Franco Valzania, Rosario Pascarella

**Affiliations:** 1Neurology Unit, Stroke Unit, AUSL-IRCCS di Reggio Emilia, Via Amendola 2, 42122 Reggio Emilia, Italy; 2Neuroradiology Unit, AUSL-IRCCS di Reggio Emilia, Via Amendola 2, 42122 Reggio Emilia, Italy

**Keywords:** lacunar stroke, perfusion, SVD, small vessel disease, MRI, CT perfusion, DSC-MRI, perforating artery, core, penumbra, acute stroke

## Abstract

The pathophysiology of lacunar infarction is an evolving and debated field, where relevant information comes from histopathology, old anatomical studies and animal models. Only in the last years, have neuroimaging techniques allowed a sufficient resolution to directly or indirectly assess the dynamic evolution of small vessel occlusion and to formulate hypotheses about the tissue status and the mechanisms of damage. The core–penumbra concept was extensively explored in large vessel occlusions (LVOs) both from the experimental and clinical point of view. Then, the perfusion thresholds on one side and the neuroimaging techniques studying the perfusion of brain tissue were focused and optimized for LVOs. The presence of a perfusion deficit in the territory of a single small perforating artery was negated for years until the recent proposal of the existence of a perfusion defect in a subgroup of lacunar infarcts by using magnetic resonance imaging (MRI). This last finding opens pathophysiological hypotheses and triggers a neurovascular multidisciplinary reasoning about how to image this perfusion deficit in the acute phase in particular. The aim of this review is to summarize the pathophysiological issues and the application of the core–penumbra hypothesis to lacunar stroke.

## 1. Introduction

The hypothesis underlying hyperacute stroke treatment both in the early and late time window is the presence of a vessel occlusion causing an hypoperfusion in the supplied territory for a variable time and with different degrees, ranging from salvageable tissue (penumbra) to irreversibly damaged tissue (core). The core–penumbra hypothesis was elaborated and proposed about 40 years ago [1]. The proposal of acute stroke as a dynamically evolving process with different thresholds of cerebral blood flow (CBF) for reversible and irreversible cerebral damage was the basis of modern vascular neurology. Subsequent evolution has led to the possibility of studying and imaging the perfusion status of brain tissue subjected to acute ischemic damage, transforming the concept of time thresholds for treatment into that of perfusion thresholds for treatment, in a relatively time-independent manner and with a progressive widening of the indications for reperfusion treatment, both intravenously (IVT) and endovascularly (EVT) [2,3,4,5].

Advanced neuroimaging techniques have been increasingly included in stroke management pathways and therefore practically used, relying on their estimation of the core–penumbra ratio or mismatch for selecting patients for treatment. Both computed tomography (CT)-based and magnetic resonance imaging (MRI)-based protocols have been used in several studies and in real practice, with CT-based advanced neuroimaging generally being more widely accessible than MR-based neuroimaging for the assessment of acute stroke. In general, advanced neuroimaging techniques use a sequentially angiographic approach (CT angiography, MR angiography, digital subtraction angiography) for assessing the presence of a large vessel occlusion (LVO) and/or a distal vessel occlusion (DVO) amenable for endovascular treatment, and a perfusion approach to evaluate tissue viability in the corresponding territory, particularly in the late time window. The quantification of core–penumbra mismatch is usually performed by using automated processing software with predefined perfusion thresholds [4]. The presence of a penumbra presupposes that there is collateral circulation capable of maintaining the CBF above the threshold of irreversible ischemic damage for a variable time.

Since the first proposal of the penumbra concept, its application has been directed almost exclusively at the LVO model and this has resulted in the fact that both neuroimaging techniques and perfusion thresholds as well as the postprocessing software have been optimized for this setting. The occlusion of a small vessel, which is the primum movens of a lacunar stroke, has been substantially neglected by this model. For a long time, it was assumed that, since it is the occlusion of a terminal artery, without the possibility of compensation by collaterals, there is no pathophysiological possibility of a penumbra and its subsequent identification. This assumption is only partially true and has been redefined over the years, leading to the study of the perfusion status in lacunar stroke and to a better understanding of the pathophysiology of this stroke subtype. In this review, we deal, in a pragmatic way, with the perfusion aspects of lacunar stroke and their impact both on the knowledge of the physiopathology of this widespread and neglected subtype of stroke and on therapeutic choices in the acute phase.

## 2. The Core–Penumbra Concept

Before introduction of the penumbra concept, i.e., a dynamically progressive cerebral ischemia over an uncertain and variable period of time, the common thought was the immediate death of brain tissue after stroke onset. Lindsay Symon [6] was the first to talk about the penumbra as the hypoperfused area around an irreversibly damaged “core” of brain tissue; he applied the astronomical term for the shadow generated by a partial solar eclipse to cerebrovascular pathophysiology. Shortly after, some animal experiments demonstrated the time course of tissue death dependent on the perfusion thresholds in primates [7] and the relationship between thresholds of cerebral ischemia and different timeframes of vessel occlusion [8].

The following step was the demonstration that increased oxygen extraction fraction marked potentially salvageable tissue using positron emission tomography (PET) [9,10]. Studies using PET introduced concepts such as “misery perfusion” as the inverse of the concept of “luxury perfusion” [11,12,13]. Finally, there was the first demonstration in humans that penumbral salvage directly predicts neurological recovery [14]. For the first time, the reversibility of brain ischemia was demonstrated and, starting from this possibility, several attempts have been made to recanalize the occluded vessel and to reflow the brain tissue, with parallel growth in neuroimaging techniques for assessing the vessel patency and perfusion status of infarcted tissue, both by using the CT-based techniques and MRI-based techniques. The core–penumbra concept finally meets the collaterals’ issue; previously, attention was almost exclusively focused on the occlusive pattern of large vessels.

Coming back to a metabolic point of view, the brain accounts for only 2% of body mass, but it consumes approximately 20% of the body’s total basal oxygen consumption at rest. The high-energy need of the brain is the reason why a sufficient blood supply has to be maintained in all circumstances and it is supplied by 16% of the cardiac blood output. The mean number of neurons of a normal adult male’s brain is 130 billion (21.5 billion in the neocortex) [15]. The brain’s oxygen consumption is mainly due to the oxidative metabolism of glucose for basic processes of a cell’s life [16]. Only synapse-rich regions may have an increase in glucose consumption (and, therefore, regional blood flow) evoked by functional activation by an order of magnitude related to the frequency of action potentials in the afferent pathways. Overall, signaling requires 87% of the total energy, mainly for the propagation of action potential and postsynaptic ATP-dependent ion exchanges: the maintenance of the membrane resting potential uses the remaining 13% [17].

The different energy requirements lead to different thresholds of energy consumption and, consequently, blood flow required for preservation of neuronal function and morphological integrity. Two critical levels of decreased perfusion were demonstrated in experimental studies: (1) a flow threshold for reversible failure of cellular functions (functional threshold); (2) a lower threshold below which the membrane processes are irreversibly lost and morphological damage occurs as infarction [1]. The range of perfusion values between thresholds (1) and (2) is the “ischemic penumbra” [7]. The main feature of the penumbra is the absence of permanent functional and morphological damage if local blood flow is restored at a sufficient level in a timely manner [18]. This functional threshold was demonstrated in nonhuman primates with an induced ischemia, where the neurological deficit gradually developed and progressed from mild paresis at 22 mL/100 g/min to complete paralysis at 8 mL/100 g/min [8]. Similarly, the registration of the electrocorticogram and evoked potentials (EPs) showed an abolition of the electric signal at 15–20 mL/100 g/min [19,20] and the disappearance of the spontaneous activity of cortical neurons around 18 mL/100 g/min. One of the most interesting findings is the existence of a large variability in the functional thresholds among individual neurons (range from 6 to 22 mL/100 g/min) [21], indicating selective vulnerability even within small cortical sectors. Moreover, the development of irreversible morphological damage is time-dependent, but this time is highly variable and location- and function-related within the brain. Animal models of middle cerebral artery (MCA) stroke provided a study of cortical function by measuring EP. Outside MCA territory, EPs are abolished even at nearly normal cortical blood flow values, thus indicating an ischemic effect on afferent pathways in the white matter [22]. Therefore, EP blockage was caused by ischemia around 6.4–7.6 mL/100 g/min in the afferent pathways and was reverted upon restoring the blood flow, demonstrating that the white matter can tolerate longer periods of severe ischemia than the cortex. Moreover, there could be a functional impairment not caused by regional ischemia but by deafferentation, and this last one has a better prognosis than cortical damage caused by direct cortical ischemia. Based on these experimental data, a discriminant curve was proposed to represent the several scenarios of residual blood flow and duration of ischemia still permitting neuronal recovery. For example, blood flow rates of almost 0, 10 or 15 mL/100 g/min lasting 25, 40 and 80 min, respectively, were the upper limit. Conversely, low flow conditions (between 17 and 18 mL/100 g/min) have a duration of ischemia tending to infinity, meaning that a very long time is needed to determine morphological damage [23]. The concept of the penumbra has been broadened and it is a highly dynamic and locally variable process, with the selective threshold for irreversible loss of function being very different for different neurons. Sometimes, critical but primarily not harmful flow decrease may trigger a dynamic process potentially leading to delayed neuronal death because of selective vulnerability [24]. Moreover, different cellular functions require different blood flow thresholds. The mean thresholds for ATP depletion (18.5 mL/100 g/min) and spontaneous neuronal activity (18 mL/100 g/min) correspond [25]. Conversely, normal protein synthesis requires a significantly higher threshold (mean value, 55 mL/100 g/min). Reduced protein synthesis, which to some extent occurs in the penumbra too, may be important for mechanisms of neuroplasticity and responsible for the deactivation of large portions of the brain outside the ischemic region itself [26,27].

Based on the threshold concept of brain ischemia, the penumbra can be measured and localized using predefined flow thresholds on quantitative flow images. In experimental studies, this concept has been applied by using a direct imaging approach of threshold-dependent biochemical disturbances on brain slices. This approach allows for distinguishing the mismatch between processes that occur only in the infarct core and others that also affect the penumbra. The most reliable way to localize the infarct core is the loss of ATP on bioluminescent images of brain tissue. Tissue acidosis is a biochemical marker of core plus penumbra. The penumbra is the difference between the respective lesion areas. The inhibition of protein synthesis starts at a higher flow level [28].

As time goes on and the brain remains hypoperfused, the infarct progresses through three phases [29] (Table 1).

In animal experiments, 3 h and 6–8 h after MCA occlusion, respectively, more than 50% and almost the entire penumbra became part of the irreversibly damaged core. However, some small regions characterized by reduced blood flow and preserved oxygen consumption could be observed for up to 24 h or even more around the lesion at the border zone of the ischemia [27,30].

Year by year, the development of treatment options for acute stroke led to cancel pathophysiological studies on the evolution of the core–penumbra relationship. Concurrently, metabolic imaging techniques have given way to morphological imaging, CT perfusion (CTP) and perfusion MRI. The concept of the penumbra in its practical application to the management of the hyperacute phase of stroke has also changed. Nowadays, the penumbra is generally considered in a dynamic approach and it associates time-sensitive hypoperfused brain tissue with decreased oxygen and glucose availability, still salvageable by intervention, restoring the blood flow; therefore, the penumbra is the potential target for neuroprotection in focal stroke [31]. Moreover, it is becoming more and more evident that the penumbra is not a single entity, but a spatially and timely dynamic and heterogeneous process, coming back to the first documentation of the heterogeneous cellular and molecular mechanisms that determine tissue fate [29]. The introduction of the conceptual framework for the neuro–glial–vascular unit (NVU) and the associated blood–brain barrier (BBB) damage helped to overcome the dichotomy of blood vessels and cells of the brain as distinct and separate entities [32,33].

The revisited concept of penumbra fate [29] started from the original notion (occlusion of a brain-supplying artery produces a central core of tissue injury destined for infarction that is surrounded by a penumbra of metabolically metastable tissue with the potential for full recovery), adding a relevant refinement. Indeed, in the early minutes and hours after ischemia onset, the core contains “pockets of injury” (“mini-cores”), surrounded by “mini-penumbras”, jeopardizing the overall hypoperfused area. Furthermore, following this hypothesis, the NVUs in the “mini-cores” have been irreversibly damaged because of flow cessation and its consequences, whereas the NVUs in the “mini-penumbras” are at least partially viable. Moreover, separate networks of microvessels, neurons, and glia associate “mini-cores” with “mini-penumbras”. Accordingly with this model, these mini-cores can grow into their respective mini-penumbras to encompass a larger region of injury. Focusing attention on the NVUs means to hypothesize that injury evolution may occur at different speeds within different “mini-cores”/“mini-penumbras”.

All this information helps to define the concept of the penumbra at the tissue level and the information deriving from neuroimaging investigations substantially represent a surrogate of what happens at the cellular level or of the measurable metabolism in selective areas of the brain using PET techniques. However, the focus of the core–penumbra model is stroke due to LVO and the involvement of a large volume of brain tissue. The transferability of these concepts, including blood flow thresholds and timing, to different subtypes of stroke, particularly small vessel disease-related stroke, is not immediate.

## 3. Definition of Lacunar Stroke

Lacunar stroke accounts for 25% of ischemic strokes [34]. It is a subtype of stroke, involving the subcortical structures, whose definition is mediated from neuropathological evidence, and its application to acute stroke imaging techniques has raised some criticism. Recent small subcortical infarctions and lacunes are partially overlapping concepts ranging from the acute phase to chronic evolution. That being said, the definition of lacunar infarcts is still highly variable among neurologists and vascular neurologists [35]. The ambiguity in terminology originates from the use of the term “lacune” (“lacuna” in Latin, meaning “hole” in English) by French neurologists in the nineteenth century for describing the presence of small cavities in the brain in pathological examination [36,37,38]. In the 1960s, C.M. Fisher described the “lacunar hypothesis” [39] starting from neuropathological observation of the presence of non-atheromasic lesions of the vascular wall in small vessels, potentially leading to the occlusion of a single long perforating artery. The subsequent damage was a small deep infarct and corresponding lacunar syndromes were described [40,41]. The introduction and implementation of neuroimaging techniques allowed for refining the concept of lacunar stroke/infarct and its differential diagnosis. Actually, lacunar infarcts are considered as small (2 to 15 mm in diameter) noncortical infarcts caused by occlusion of a single penetrating branch of a large cerebral artery [42,43]. These branches arise at acute angles from the large arteries of the circle of Willis, stem of the MCA, or the basilar artery. Although this definition implies that pathological confirmation is necessary, in vivo diagnosis may be made by matching the presence of appropriate clinical syndromes with radiological tests. Not all small deep infarcts are lacunar, and the diagnosis of lacunar infarct also requires the exclusion of other etiologies of ischemic stroke. It is noteworthy that pathology studies defining lacunar infarcts were performed in the chronic phase of stroke [42]; some neuroimaging studies in the acute phase (<10 days from stroke onset) have used 20 mm or even 25 mm on DWI [44,45] as the upper size limit for lacunar infarct, since some volume reduction is expected over time. RSSIs have been imaged using both MRI and CT scan, MRI being more sensitive, particularly in the hyperacute phase. A comparative study between CT and MRI within 6 h from symptom onset showed that among 15/64 patients diagnosed with RSSI based on clinical/CT criteria, 10/15 (67%) had a final MR-DWI-confirmed diagnosis of RSSI [46]. Finally, the lacunar syndrome correlated with a clinical lacunar infarction in only 50% to 60% of the cases using MRI [47,48,49].

A summary of the definitions is provided in Table 2 and, for the purpose of this review, we consider only pure SVD-related lacunar infarcts.

Another source of variability comes from the etiological classifications of ischemic stroke, where a mixture of clinical, neuroimaging and histopathological issues are combined [53]. The Trial of Org 10,172 in Acute Stroke Treatment (TOAST) classification [54] is one of the most widely used and contains a category for SVD-related stroke. In this category, the definition is mainly clinical, referring to the five most known of the 21 different lacunar syndromes described by Miller-Fisher [42]. The hallmark clinical feature is the absence of cortical signs or symptoms (but thalamic aphasia and neglect are late valuable exceptions) [55,56,57,58] and the corresponding neuroimaging using CT or MRI could be normal or show a relevant brain stem or subcortical lesion with an axial diameter < 15 mm. Moreover, any potential embolic source or stenosis >50% in an ipsilateral proximal artery should be excluded. The Causative Classification of Stroke (CCS) was based on a different reasoning, providing a web and evidence-based algorithm for causative and phenotypic stroke subtypes [59]. The algorithm led to the definition of a lacunar stroke when there is an acute ischemic lesion < 20 mm at the greatest diameter in the territory of a penetrating artery and without focal pathology in the parent artery. The ASCO (A: atherosclerosis; S: small-vessel disease; C: cardiac pathology; O: other causes) and ASCOD (D: dissection) [60] classifications proposed a graded causality for each category, including SVD. Lacunar stroke is considered potentially causal when there is a small, deep infarct < 15 mm in diameter using neuroimaging in the territory corresponding to symptoms, accompanied by other SVD-related neuroimaging findings [52].

Moreover, there is a histopathological definition, and it has many potential pathophysiological implications, which are detailed in the next sections.

## 4. Histopathological Lesson on Lacunar Infarcts

The most used and convincing histopathological definition is provided by Poirier and Derouesné [61]: (1) type Ia is an old infarct <20 mm with pan-necrotic cavitation and scattered macrophages; (2) type Ib is an incomplete infarct with selective loss of vulnerable elements. Moreover, histopathology provides interesting information not only about the infarcted tissue but also about the occluded vessel in lacunar infarct.

C.M. Fisher was the first to note that most lacunes were distal to occlusive lesions of small perforating arteries [39,62,63,64,65,66]. The vessel occlusive lesions include arteriosclerosis/atherosclerosis, arteriolosclerosis and lipohyalinosis [67,68]. The main vessel lesions are summarized in Table 3.

Lipohyalinosis is the major driver of SVD, and it is typical of lacunar stroke. Miller-Fisher proposed the definition of “hypertensive cerebral vasculopathy”, describing it as occlusion of the lumen of an artery, usually <200 mcm in diameter, whose wall structure is destroyed and highly fragmented, surrounded by hemosiderin-filled macrophages. The reorganization of a small perforating branch occlusion may include retrograde occlusion (e.g., in pontine penetrators with normal restoration of flow at the junction of the parent artery) and the formation of small “channels of recanalization”. Finally, a histopathological progression of lipohyalinosis encompasses three stages [62]: (1) preferential involvement of the basal ganglia; (2) involvement of the deep white matter, the cerebellum, the thalamus, and finally the leptomeningeal arteries; (3) extension to the brainstem.

The histopathology provides relevant information but its main limitation for inferring the mechanisms of acute lacunar infarcts is due to the fact that autopsy studied old, affected vessels [70]. Conversely, neuroimaging techniques are of paramount importance to increasing understanding of acute and chronic SVD, but do not directly visualize small vessels. Actually, the two main mechanisms involved in the pathophysiology of SVD and lacunar infarct are thought to be endothelial dysfunction and BBB disruption [71]. Endothelial dysfunction is responsible for vasoconstriction with impaired autoregulation, mainly driven by local blood pressure, leading to an inability to maintain perfusion distally [51]. Moreover, high arterial pressure has been proposed to degrade the BBB, resulting in local edema and plasma protein deposition in the vessel wall and triggering wall damage [72].

The histopathology of brain tissue provides different and complementary information. The findings of Miller-Fisher’s examination are that chronic lacunes had trabeculated cavities surrounded by gliosis and few macrophages, while recent lacunes contained numerous macrophages and had minimal gliosis. The wide range of histopathological morphology solicited the proposal of a classification schema in 1984 [61]: (1) type I lacunes are ischemic lesions; (2) type II are microbleeds where the lacunar cavity is filled with hemosiderin-laden macrophages; (3) type III are dilated perivascular spaces. Type I lesions are characterized by an irregular cavity ranging 1–20 mm and the most common locations were the putamen, caudate, thalamus, pons, internal capsule and hemispheric white matter [73]. They were additionally parted in two subtypes: (1) type Ia lesions are old infarcts showing cavitation and surrounded by rare, scattered macrophages; (2) type Ib lesions are seemingly incomplete infarcts where a selective loss of the most vulnerable cells has occurred [74]. The study by Lammie et al. on 172 autopsy brains addressed the issue of type Ib lesions (incomplete infarcts) and found them in association with type Ia infarcts [74]. The conclusion of the authors was that each type of lesion is a step in a wider spectrum, proposing three levels. The histopathology of type Ib lacunar lesions is summarized in Table 4.

The etiology of type Ib parenchymal lesions has been a matter of debate because it has implications for therapeutic targets. Some authors proposed that type Ib lesions are secondary to local edema instead of ischemia [75], describing them as “edema-associated gliosis” [76]. Notably, similar descriptions have been proposed in animal models [77,78,79]. This description recalls one of the first histopathological definitions by Pierre Marie in 1901 [38,80], distinguishing between “etat lacunaire” and “etat criblé”, caused, respectively, by a “local arteriosclerotic process” [81] and “destructive vaginalitis”, in which perivascular spaces demolished adjacent areas of brain tissue [80]. In the description of lipohyalinosis by C.M. Fisher [62], the mechanism of destructive vaginalitis recurs, in which the vessel showed patent dilation of the perivascular space and destruction of the surrounding parenchyma (similar to type III lesions). It is also possible that ischemia is the main underlying mechanism for type Ib lesions, similar to type Ia [82]. One of the most convincing hypotheses is that these incomplete type Ib lesions differ from type Ia lesions secondary to a less severe degree and a shorter duration of ischemia, with a different etiology of ischemia being less likely. Moreover, a combination of the two hypotheses (edema and ischemia) cannot be excluded.

In lacunar infarct, the process of injury (ischemia or infarct), repair and recovery are not static but highly dynamic. When the brain parenchyma loses its blood supply to a degree that affects its functioning but does not cause necrosis (as in the penumbra), an ischemia occurs. On the contrary, the parenchyma, when apoptosis and necrosis occur, is called an infarct. Ischemia and infarct represent the ends of a spectrum, balanced by continuous blood flow changes. There is increasing evidence supporting the role of decreased blood flow in lacunes, small subcortical infarcts and white matter hyperintensities (WMHs), as already known in large vessel occlusions, but with different features. A recent study showed decreased CBF and vascular reactivity in WMHs versus homologous contralateral normal-appearing white matter [83]. The decrease in CBF in WMHs over time [84] and low baseline CBF were associated with the development of new WMHs [85] in a follow-up of 4 years and 18 months, respectively. However, the hypothesis of lower CBF preceding the development of lacunar stroke and WMH is debated [86]. Another theory postulates the role of chronic hypoperfusion in the progression or enlargement of lacunes. A histopathological study [87] may support this mechanism, being the periinfarct region characterized by axon disorganization up to 150% of the infarct diameter with markers of injury to myelin out of proportion to axons.

Finally, the repair process may provide some useful information, but it is not easy to study it in vivo outside of animal models. Serial neuroimaging studies [88,89,90] have explored the fate of lacunar infarcts and WMHs, finding that some RSSIs may become undetectable and some WMHs can regress on repeat imaging. One of the proposed mechanisms is the local alteration in blood flow with a WMH “penumbra” hypothesis [43] underlying the predilection of incident lacunes for the edge of WMHs.

It is possible that a combination of involvement in lower-order branched vessels and progressive worsening of vessel wall disease/increasing number of involved vessels creates flow-limiting areas of vessel disease contributing to downstream hypoperfusion and WMH or lacune evolution [43].

## 5. Pathophysiology of Perfusion Issues in Lacunar Stroke

Small subcortical infarct is caused by the occlusion of a chronically and progressively thickened small perforating artery or by plaque in the parent artery [71,91]. According to the first proposed pathophysiological hypothesis, this occlusion provokes decreased perfusion downstream because of the limited collateral capacity of penetrating arteries. This hypothesis has been recently refined due to the demonstration that anterograde and retrograde collaterals may play a critical role in maintaining cerebral perfusion in RSSI and perfusion compromise was closely associated with final infarct size [92]. Indeed, 52.2% to 76% of patients with single subcortical infarct had cerebral perfusion deficits within 24 h of symptom onset as shown using an MRI [92,93,94]. Moreover, the occurrence of an occlusion in the collateral circulation because of the sludging of blood cells in the hypoperfused distal branches affects the expansion of the ischemic lesion and the progression of neurological deficit [95]. Using dynamic contrast-enhanced (DCE) MRI, patients with hypoperfusion patterns had the highest rate of early neurological deterioration [92]. Other issues in the pathophysiology of RSSI are endothelial dysfunction and blood–brain barrier leakage [51,96,97]. Indeed, as previously detailed, a nonocclusive mechanism has been postulated in a subtype of RSSI, i.e., perivascular edema-mediated lesions, in which there is no occlusion but the tissue around the vessel is damaged by edema fluid [74]. This may suggest an important role for edema-mediated tissue damage, irrespective of the presence of a permanent vessel occlusion. Moreover, it may represent the histopathological appearance of chronic hypoperfusion in a small area of incomplete infarction because of a temporary or less severe ischemia. Another potential pathophysiological mechanism underlying ischemic lesion growth beyond the hypoperfused area in small subcortical infarction is cytotoxic mechanisms at the border of the tissue damage [98]. In this subtype of RSSI (type Ib lacunes or “incomplete lacunar infarcts”), we can expect none or only small perfusion deficit. Doege et al. [99] reported an “inverse mismatch” (DWI > perfusion-weighted imaging or PWI) in 6 patients with RSSI and a further increase in DWI lesion size as well as perfusion deficit within the following 24 h. Fiebach [98] divided RSSI patients into three groups based on initial imaging findings: no mismatch (initial DWI lesion and mean transit time (MTT) deficit volumes differed by less than 20%) for 3/19 patients, positive mismatch (the initial MTT deficit exceeded the initial DWI lesion by more than 20%) for 2/19 patients and inverse mismatch (initial DWI at least 20% larger than initial MTT) in 14/19 patients. The early reperfusion hypothesis because of early spontaneous recanalization is not a convincing explanation for the inverse mismatch pattern. The apparent diffusion coefficient (ADC) decrease and later infarction suggest that cerebral areas became severely damaged without an underlying perfusion drop (normal MTT) in the negative mismatch zone, leaving room the spread of depression-like events induced by cytotoxic substances. Thus, a reasonable explanation for the inverse mismatch may be that the central ischemia-induced lesion leads to perilesional cellular damage due to the release of cytotoxic agents and perfusion may still be intact in the affected areas. This hypothesis would also explain the large number of patients with a PWI deficit smaller than the ischemic lesion using DWI [93]. Another explanation might be the partial recanalization of a previously occluded perforating artery or collateral blood flow from adjacent vascular territories.

Collateral flow plays a relevant role in RSSI too. Indeed, if, in the past, lacunar infarction was thought to be caused by the occlusion of a terminal penetrating artery, without collateral circulation and the occurrence of penumbra [100,101], on the contrary, postmortem studies of the human brain revealed anastomoses of the major perforators and precapillary arterioles [102,103]. The first documentation of anterograde and retrograde collaterals in acute lacunar infarction was described by Forster et al. using DCE-MRI [93]. CTP was useful for demonstrating a delayed compensatory collateral supply in recent SSI [104]. A further issue to support the penumbra hypothesis in lacunar stroke is the documentation that stroke patients with lacunar infarcts may also benefit from pharmacological reperfusion [105]. Huang et al. [92], in 103 patients with RSSI undergoing DCE-MRI within 24 h of stroke onset, identified three perfusion patterns: normal perfusion in 25 (24%), compensated perfusion in 31 (30%), and hypoperfusion in 47 (46%). This study suggests that anterograde collaterals may play a crucial role in maintaining the adequacy of microvascular perfusion after an RSSI because patients with the hypoperfusion pattern had lower rates of anterograde collaterals. Anterograde collaterals may come directly from partially occluded arteries or indirectly from anastomosis of the proximal penetrating arteries, but the first case is more likely because of the fewer proximal anastomoses from basilar arteries and MCA [102]. The adequacy of the anterograde collaterals may explain why one quarter of patients had normal perfusion patterns in the infarct areas.

It is possible that a perfusion evaluation distinguishing the penumbra from the core is not possible for RSSI in the same way it is performed for small and large territorial infarctions. The resolution of CTP maps does not allow for this task, as well described in Garcia-Esperon 2021 [106], where, in the lacunar group, the median core volume was 0 (0–0.9) mL (IQR), the median penumbra volume was 0.4 (0–2.9) mL (IQR) and the median hypoperfused lesion volume was 0.7 (0–4.6) mL (IQR). These measures correspond to a median volume in DWI of 1.9 (0.8–5.5) mL (IQR), so DWI hyperintensity size is greater than the size of hypoperfused tissue in CTP. Indeed, small infarcts can be omitted even when they are included in the interrogated volume of CTP because CTP maps have relatively low spatial resolution. Areas of chronic infarction can be confusing in CTP, although they are generally obvious on unenhanced CT. Most of the tissues with chronic infarction show a low but persistent degree of metabolism and decreased but measurable perfusion parameters [107]. Another explanation for lacunar lesions without or with only small perfusion deficit might be perivascular edema-mediated lesions, in which there is no occlusion but the tissue around the vessel is damaged by edema fluid [74]. It is obviously hard to translate the evidence from histopathology to neuroimaging appearance [74] and the proposed patients had several different diseases directly or indirectly affecting the brain as a potential confounder. The proposed issue of type Ib lacunes or “incomplete lacunar infarcts”, to distinguish them from classical type Ia lacunes [61], consists of a small area of perivascular rarefaction and selective neuronal loss with or without associated astroglia response and without a clear relation to an occluded perforating vessel. The morphology of the “incomplete lacunar infarct” is the same as that described by Ma and Olsson [75] as “oedema-related gliosis”. This may suggest an important role for oedema-mediated tissue damage, whether or not this is associated with permanent vessel occlusion, and it may represent the histopathological appearance of chronic hypoperfusion in a small area of incomplete infarction, implying temporary or less severe ischemia. Another possibility is the existence of a different underlying pathophysiological mechanism in small subcortical infarction with ischemic lesion growth beyond the hypoperfused area possibly induced by cytotoxic mechanisms at the border of the tissue damage [98]. Previously, only a few MRI studies addressed the perfusion of RSSI [99,108,109]. In the series by Gerraty et al. [108], 10/17 patients with RSSI had a perfusion deficit smaller than the DWI lesion; in two cases, the infarct spread beyond the initial extent in the absence of any perfusion deficit. As previously said, Doege et al. [99] reported an “inverse mismatch” (DWI > PI) in 6 patients with RSSI with a further increase in DWI lesion size and perfusion deficit within 24 h. The final infarct size after 1 week was larger than the hypoperfused area both in the acute phase and at the follow-up examination [99]. In Fiebach et al. [98], the initial DWI lesion had a mean volume of 1.82 mL (SD 1.2 mL) and a mean perfusion deficit on MTT maps of 0.72 mL (SD 0.69 mL). The size difference between DWI and MTT lesions was highly significant (*p* = 0.002), confirming an inverse mismatch between hypoperfusion and infarct. On day 2, mean DWI lesion volume increased to 2.78 mL (SD 1.39 mL) and inverse mismatch was evident in all patients (*p* < 0.0001). Final lesion volume (FLV) at day 6 ranged from 0.64 to 6.0 mL (mean 3.2 mL; SD 1.6 mL) and it was a median 1.69 times larger than the initial DWI lesion and 3.63 times larger than the initial MTT lesion (mean values 2.33 and 5.23, respectively). Compared with final infarct volume, the initial DWI lesion was 55.7 FLV% (SD 34.3%) and the mean initial MTT deficit was 24.7 FLV% (SD 21.6%). Contrary to lesion growth using DWI, the MTT deficit showed no change (*p* = 0.98) from day 1 (24.7 FLV%) to day 2 (23.6 FLV%).

Rudilosso et al. [104] provided interesting information about the pathophysiology of perfusion abnormalities in RSSI by using CTP. In a visual evaluation, CBF and CBV were classified as reduced, normal or increased, whereas time to drain (TTD) was classified as normal or delayed in comparison with the corresponding contralateral area. Then, the perfusion pattern was rated as follows: hypoperfusion (reduced CBF and delayed TTD), normoperfusion (not altered CBF and TTD) and hyperperfusion (increased CBF with normal TTD). Not surprisingly, CBF and CBV values were higher in patients with hyperperfusion, while TTD was longer in patients with hypoperfusion. Patients with hyperperfusion or normal perfusion were more likely to show an acute hypodense lesion at the baseline non-contrast CT (NCCT). Yamada [110] reported that a higher MTT ratio (>1.26) and a lower CBF ratio (<0.76) on perfusion CT scans can predict progressive lacunar infarction in the lenticulostriate territory.

In general, although there were technical differences in the individual studies, time-based maps were found to be more sensitive than flow-based maps.

## 6. Practical Implications and Perspectives

The perfusion techniques in the hyperacute phase of stroke were studied and their scope of application and thresholds aimed at identifying patients amenable to IVT and EVT treatment, overcoming the concept of time and starting from physiopathology of the ischemic damage of the brain tissue involved. As previously said, the premise of these studies is the presence of a territorial ischemia conditioned by an LVO or DVO. Lacunar stroke, being caused by the occlusion of a single small perforating artery or by atherothrombotic involvement of the corresponding parent artery, is outside this scope, so there is no reliable information on the use of perfusion techniques in the hyperacute phase for the identification of a mismatch in the territory of a perforating artery.

Indeed, lacunar perfusion lesions tend not to be visible on postprocessed core–penumbra maps, mainly because core–penumbra maps are typically smoothed on automated software including only relatively large groups of hypoperfused pixels in the core–penumbra lesion. Previous small studies suggested that visual assessment of the MTT map had reasonable sensitivity for detecting lacunar infarction [111,112,113,114]. DWI-MRI has excellent sensitivity for lacunar stroke but is not readily accessible for acute stroke patients. CT perfusion is more easily accessible than MRI in several settings, but lacunar stroke accounts for most of the 50% rate of false negatives [115,116]. A recent systematic review on the topic of CTP in acute lacunar stroke [117] showed several sources of heterogeneity in the global performance of this technique. In particular, there are issues related to the technique (scanner features and technological era, rows number, perfusion protocol, vertical coverage, postprocessing software versus visual assessment of the maps) but also issues related to the disease and the patient (size and location of lacunar infarcts, even excluding infratentorial stroke; administration of IVT; etc.).

Moreover, small perfusion defects seen in CTP and perfusion MRI, indicating the existence of penumbra, were associated with the development of early neurological deterioration (END) in a few series [92,110,118]. However, whether the penumbra hypothesis could be applied to microcirculation is still a matter of discussion, but when starting from the assumption of the validity of this hypothesis, collateral circulation may be considered as playing a critical role in maintaining perfusion in the penumbral region of RSSI. The first documentation of anterograde and retrograde collaterals in acute lacunar infarction was described by Forster et al. using DSC-MRI [93]. CTP was useful for demonstrating a delayed compensatory collateral supply in RSSI [104]. A further issue in supporting the penumbra hypothesis in lacunar stroke is the documentation that stroke patients with lacunar infarcts may also benefit from pharmacological reperfusion [119]. Finally, the hemodynamic changes and collaterals of RSSI were evaluated in a couple of studies using DSC-MRI [92,93]. Huang et al. [92] recruited 103 patients with acute SSI in penetrating artery territories and performed MRI within 24 h of stroke onset. Three perfusion patterns were identified: (1) normal perfusion in 25 (24%); (2) compensated perfusion in 31 (30%); (3) hypoperfusion in 47 (46%). Moreover, the authors evaluated the development of anterograde (defined as when the filling contrast appeared inside) or retrograde (defined as when the contrast filled from outside to inside in continuous images) collaterals. Patients with normal perfusion patterns were classified as having both anterograde and retrograde collaterals because the cerebral perfusion was similar to the contralateral side. Patients with a hypoperfusion pattern had the highest rate of END (32%, *p* = 0.007), the largest initial and final infarct volumes (*p* < 0.001 and *p* = 0.029), the lowest relative CBF (0.63, *p* < 0.001), and the lowest rate of anterograde and retrograde collaterals (19%, *p* < 0.001; 66%, *p* = 0.002). Anterograde collaterals were associated with higher relative CBV (0.91 vs. 0.77; *p* = 0.024) and a higher rate of deep cerebral microbleeds (48 vs. 21%; *p* = 0.028), whereas retrograde collaterals were associated with higher systolic and diastolic blood pressure (*p* = 0.031 and 0.020), smaller initial infarct volume (0.81 vs. 1.34 mL; *p* = 0.031) and a higher rate of lobar cerebral microbleeds (30 vs. 0%; *p* = 0.013). These findings suggest that anterograde collaterals play a main role in maintaining adequate microvascular perfusion after RSSI. Anterograde collaterals may come directly from partially occluded arteries or indirectly from anastomosis of the proximal segments of penetrating arteries, but the first case is more likely because of the fewer proximal anastomoses from basilar arteries and MCA documented in anatomical studies [102]. The adequacy of the anterograde collaterals may help to explain why one quarter of patients with RSSI had normal perfusion patterns in the infarct areas. It is possible that the absence of anterograde collaterals increases autoregulation through the elevation of blood pressure, which may increase the retrograde collaterals [120,121].

The impact of the identification of a perfusion deficit in CTP in the acute phase of stroke is one of the treatment decisions, in particular IVT. Conversely, in the late window, the lack of identification of a perfusion deficit usually does not allow for the performance of IVT. As detailed in the systematic review on CTP in lacunar stroke patients [117] and in the present review, the pathophysiology of RSSI and technical issues of available techniques may affect the possibility to image the perfusion status of these patients and the best treatment has not been defined. After the acute phase, another matter is the etiology of RSSI, because pure SVD mechanisms are the main causes, but—particularly in larger lesions, where a putative involvement of more than one penetrating artery can be supposed—atherothrombosis of the parent vessel and, less frequently, embolic causes also have to be considered and investigated. The recent update to the STRIVE criteria [122] deeply addressed this issue.

## 7. Conclusions

The pathophysiology of lacunar stroke is complex and involves collaterals and perfusion status with penumbra at a smaller scale than stroke due to LVO. Hemodynamic factors play a role in the occurrence and evolution of RSSI and WMH and the exact mechanisms are not yet fully understood. Better knowledge of these issues would allow these processes to be targeted for focused therapeutic strategies.

## Figures and Tables

**Table 1 diagnostics-13-02003-t001:** The three phases of infarct progression.

Phases	Cerebral Blood Flow Threshold	Mechanisms of Damage
Acute phase (few minutes from ischemia onset)	<20% of pre-occlusion values (core)	Ischemia-induced energy failure and terminal depolarization of cell membranes
Subacute phase (from 4–6 h to >24 h in different models)	25 and 50% of pre-occlusion values (penumbra)	The irreversible damage expands into the areas around the core until after several hours, so the core expands into the penumbra, defined as areas of decreased CBF and O_2_ metabolism, but increased OEF. The main mechanism is periinfarct spreading depression, which starts at the border of the infarct core and spreads over the ipsilateral hemisphere. The metabolic rate of the tissue but not rCBF markedly increases, leading to stepwise accumulation of lactate with each depolarization, loss of ionic gradients, until cell death.
Delayed phase (several days or even weeks)	25 and 50% of pre-occlusion values (penumbra)	Secondary phenomena (vasogenic edema, inflammation, programmed cell death) may contribute to further progression of tissue damage

CBF: cerebral blood flow; OEF: oxygen extraction fraction.

**Table 2 diagnostics-13-02003-t002:** Main definitions of lacunar issues.

Issue	Definition
Lacunar syndrome	A clinical diagnosis based on the presence of pure motor, sensory or sensorimotor stroke; or ataxic hemiparesis; with acute onset, fast development, a duration of at least 24 h and with no apparent cause other than that of vascular origin [50]. Neuroimaging examination may or may not show a DWI-positive lesion in the territory of a perforating arteriole.
RSSI	It substitutes the term “lacunar infarct” and refers to neuroimaging evidence of recent infarction in the territory of a single perforating artery, seen as a hyperintense lesion on FLAIR and DWI-MRI, generally < 20 mm diameter in the axial plane, with features or clinical symptoms consistent with a lesion occurring in the previous few weeks [51].
Lacunes	The corresponding chronic phase of RSSI and defined as a neuroimaging feature identified as a round or ovoid, subcortical, fluid-filled cavity, with a CT or MRI signal similar to that of the CSF, and diameter between 3 and 15 mm, consistent with a previous RSSI or hemorrhage in the territory of a single perforating artery [51,52].

DWI: diffusion-weighted imaging; FLAIR: fluid-attenuated inversion recovery; RSSI: recent small subcortical infarct.

**Table 3 diagnostics-13-02003-t003:** The three main histopathological lesions in the perforating arteries of patients with lacunar stroke according to Miller-Fisher [39,62].

Lesions	Caliper of the Involved Vessels	Main Findings
Arteriosclerosis/atherosclerosis	200–800 mcm	It involves similar mechanisms as atherosclerosis of larger vessels with less prominent calcifications. C. M. Fisher described microatheromas in the proximal segment of perforating arteries, junctional atheromas in the origin of a perforator, or mural atheromas [62].
Arteriolosclerosis	40–150 mcm	It causes concentric hyaline thickening of the vessel wall [68].
Lipohyalinosis	40–300 mcm	This appearance is due to the aggregation of lipids and proteins [69]. It is a noninflammatory fibrinoid vessel wall necrosis, where connective tissue largely replaces the affected vessel walls until occluding the lumen. This segmental arteriolar disorganization shows three overlapping findings: vessel enlargement, hemorrhage (sometimes suggesting a microdissecting mechanism) and fibrinoid deposition. Moreover, lipohyalinosis is a focal process involving a longitudinal segment corresponding to 2–3 times the vessel diameter, mainly located at sites of bifurcation and branching [39].

**Table 4 diagnostics-13-02003-t004:** Grading of type Ib lacunar infarcts according to Lammie et al. [74] on histopathological grounds including their size, the degree of parenchymal integrity and surviving neurons and glia, and the presence of reactive glia and inflammatory cells.

Grade	Size	Main Features
Grade 1	0.5 × 0.5 mm	Parenchymal rarefaction with mild decrease in the number of neurons (occasional “ghost” neurons), normal oligodendroglia, lack of reactive astrocytes and inflammatory cells.
Grade 2	Intermediary between grade 1 and grade 3	Intermediate involvement between grade 1 and 3.
Grade 3	10 × 1.0 mm	Wide cavitation of the parenchyma with spongiosis; absence of vital neurons with rare surviving oligodendroglia. The cavities have gliovascular septa and surrounding gliosis with inflammatory cells within and around the cavities

## Data Availability

Not applicable.

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
