# Peer review of "Perfusion Status in Lacunar Stroke: A Pathophysiological Issue"

_diagnostics, 2023, doi:10.3390/diagnostics13122003_

Round 1

Reviewer 1 Report

This is a very well written review article about lacunar stroke.

Since these patients get fibrinolytic therapy in the acute stage, it would be interesting, if its impact would be addressed in the paper.

How can we differentiate from microembolic lesions?

L. 299 writing errors

L 480 DWI volume on day 2 not presented

L 812 Citation incomplete

Author Response

First of all, we would thank the reviewer for the appreciation of our paper and for the comments.

We addressed the suggestions as in the attached manuscript and, in more details:

Since these patients get fibrinolytic therapy in the acute stage, it would be interesting, if its impact would be addressed in the paper.

How can we differentiate from microembolic lesions?

We added in the last section these sentences addressing the proposed issues:

"The impact of the identification of a perfusion deficit in CTP in the acute phase of stroke is on the treatment decisions, in particular IVT. Conversely, in the late window the lack of identification of a perfusion deficit usually do not allow to perform IVT. As detailed in the systematic review on CTP in lacunar stroke patients [117] and in the present review, the pathophysiology of RSSI and technical issues of available techniques may affect the possibility to image the perfusion status of these patients and the best treatment has not been defined. After the acute phase, another matter is the etiology of RSSI, because pure SVD mechanisms are the main causes, but, in particular in larger lesions, where a putative involvement of more than one penetrating artery can be supposed, also atherothrombosis of the parent vessel and, less frequently, embolic causes have to be considered and investigated. The recent update of the STRIVE criteria [123] deeply addressed this issue."

L. 299 writing errors

Corrected

L 480 DWI volume on day 2 not presented

It is reported in line 474: "On day 2, mean DWI lesion volume increased to 2.78 ml (SD 1.39 ml)"

L 812 Citation incomplete

I completed the citation. In the meantime tha paper was published.

Reviewer 2 Report

The authors provide a thorough narrative review of current concepts regarding pathophysiology and advanced neuroimaging, incorporating perfusion techniques, in lacunar stroke. The manuscript is concise and of great interest to clinicians who deal with acute ischemic stroke patients. I have only a few suggestions that may further enhance this work , if adapted. 

There are a lot of syntax errors in some instances that may need an editing review after the acceptance and some typos. I just quote a few that came to my attention: 

Typos: raw 65 “over the years.” 

Raw 187 “metastable)?

Raw 211 “and lacune and are partially

202-204 I suggest rephrasing.

376 “being limited…” please rephrase.

388 of instead of od

389-393 “it may suggest …severe ischemia” .Please rephrase

I think that paragraph 521-557 in the practical implications and perspectives is somehow repeating the discussion of section 5 and the pathophysiology of perfusion issues in lacunar stroke. I suggest incorporating it in the previous section, which could as well be more condensed. In the practical implications and perspectives section I suggest that the authors offer some therapeutic recommendations (such as intravenous thrombolysis or considerations regarding blood pressure management in patients with persistent perfusion deficit, etc.), as well as perhaps some case studies and illustrations that highlight the perfusion patterns in lacunar strokes. It could be useful to have a table that also summarizes the perfusion patterns evaluated by several advanced neuroimaging techniques (CTP, MRI DSC, MRI DCE).

Incorporated in the previous paragraph. 

Author Response

First of all, we would thank the reviewers for the appreciation of our manuscript and for the suggestions. We tried to address them, as in the revised document.

More in detail, we changed the text according to the comments and corrected the typos: 

Typos: raw 65 “over the years.” 

Corrected

Raw 187 “metastable)?

It is a term about the state of equilibrium coming from the physics (the penumbra concept was originally derived and named from astronomy) and it means "stable provided it is subjected to no more than small disturbances". It seems to us short and informative to describe the penumbra.

Raw 211 “and lacune and are partially

Corrected

202-204 I suggest rephrasing.

Done

376 “being limited…” please rephrase.

Done: " However, the main focus of the core-penumbra model is stroke due to LVO and involving a large volume of brain tissue. The transferability of these concepts, including the blood flow thresholds and timing, to different subtypes of stroke is not immediate, in particular to small vessel disease related stroke."

388 of instead of od

Changed

389-393 “it may suggest …severe ischemia” .Please rephrase

Done: "It may suggest an important role for edema-mediated tissue damage, irrespectively from the presence of a permanent vessel occlusion. Moreover, it may represent the histopathological appearance of chronic hypoperfusion in a small area of incomplete infarction because of a temporary or less severe ischemia."

I think that paragraph 521-557 in the practical implications and perspectives is somehow repeating the discussion of section 5 and the pathophysiology of perfusion issues in lacunar stroke. I suggest incorporating it in the previous section, which could as well be more condensed. In the practical implications and perspectives section I suggest that the authors offer some therapeutic recommendations (such as intravenous thrombolysis or considerations regarding blood pressure management in patients with persistent perfusion deficit, etc.), as well as perhaps some case studies and illustrations that highlight the perfusion patterns in lacunar strokes. It could be useful to have a table that also summarizes the perfusion patterns evaluated by several advanced neuroimaging techniques (CTP, MRI DSC, MRI DCE).

We would thank the reviewer for the suggestion, and we shortened the section, as suggested, but we chosen to mantain here some information because of the need to define the main messages with a practical implication. Therapeutical implications (IVT) were added according to the suggestions. We think that addressing other issues (blood pressure management in ntha cute phase, etc) or adding examples and systematization of the findings of different perfusion tehcniques could not be evidence-based or supported and both issues are evolving fields, where no certain indications are available. Obviously, the uncertainty is a matter which deserves attention, but it  may be the topic of another review. We confirm our willingness to add these information (it means a new paragraph), if requested by the editor. This review is the theoretical framework of a systematic review on CTP in lacunar stroke we recelty published (https://www.mdpi.com/2075-4418/13/9/1564).